# Intermittent Exposure to a 16 Hz Extremely Low Frequency Pulsed Electromagnetic Field Promotes Osteogenesis In Vitro through Activating Piezo 1-Induced Ca^2+^ Influx in Osteoprogenitor Cells

**DOI:** 10.3390/jfb14030165

**Published:** 2023-03-20

**Authors:** Yangmengfan Chen, Benedikt J. Braun, Maximilian M. Menger, Michael Ronniger, Karsten Falldorf, Tina Histing, Andreas K. Nussler, Sabrina Ehnert

**Affiliations:** 1Siegfried Weller Institute at the BG Trauma Center Tübingen, Department of Trauma and Reconstructive Surgery, University of Tübingen, Schnarrenbergstr. 95, D-72076 Tübingen, Germany; 2Sachtleben GmbH, Haus Spectrum am UKE, Martinistraße 64, D-20251 Hamburg, Germany

**Keywords:** electromagnetic field, bone, piezo 1, calcium ions, osteoprogenitor cells

## Abstract

Exposure to extremely low frequency pulsed electromagnetic fields (ELF-PEMF) is supposed to simulate local EMF generated during mechanical stimulation of bone and may therefore be used to improve bone regeneration. This study aimed at optimizing the exposure strategy and investigating the underlying mechanisms of a 16 Hz ELF-PEMF, previously reported to boost osteoblast function. Comparing influences of daily continuous (30 min every 24 h) and intermittent (10 min every 8 h) exposure to the 16 Hz ELF-PEMF on osteoprogenitor cells revealed that the intermittent exposure strategy enhanced the 16 Hz ELF-PEMF effects regarding cell numbers and osteogenic function. Gene expression of *piezo 1* and related Ca^2+^ influx were significantly increased in SCP-1 cells with the daily intermittent exposure. Pharmacological inhibition of piezo 1 with Dooku 1 largely abolished the positive effect of the 16 Hz ELF-PEMF exposure on osteogenic maturation of SCP-1 cells. In summary, the intermittent exposure strategy enhanced the positive effects of 16 Hz continuous ELF-PEMF exposure in terms of cell viability and osteogenesis. This effect was shown to be mediated by an increased expression of *piezo 1* and related Ca^2+^ influx. Thus, the intermittent exposure strategy is a promising way to further optimize the therapeutic effects of the 16 Hz ELF-PEMF regarding fracture healing or osteoporosis.

## 1. Introduction

The German anatomist and surgeon Julius Wolff reported strong relations between bone remodeling and mechanical stimuli, known as “Wolff’s law” [1]. Further investigations proved the piezoelectric properties of bone that are essential for transferring mechanical stimuli into electric signals, affecting the metabolism of osteoprogenitor cells [2]. Therefore, exposure to extremely low frequency pulsed electromagnetic fields (ELF-PEMF) is a promising adjunct therapy that attempts to simulate external physical stimuli to bones by delivering electric signals to osteoprogenitor cells [3]. The frequency of the electric fields generated by the physical stimuli is one of the essential parameters that induce the biological effects. It has been reported that walking-induced mechanical forces elicit a frequency of approximately 15 Hz on the human skeletal system [4]. This is in line with a preceding study that screened different ELF-PEMFs with frequencies ranging from 10 to 90.6 Hz for their osteogenic potential. In this study, daily exposure to the ELF-PEMF with a fundamental frequency of 16 Hz most effectively induced osteoblast function [5]. The clinical use of this specific 16 Hz ELF-PEMF confirmed this finding by accelerating osseous consolidation of high tibia osteotomies [6]. Although the non-invasive treatment and therapeutic effects of the 16 Hz ELF-PEMF were proven, the challenge of optimizing the ELF-PEMF treatment setting for therapeutic use remained. When compared to other studies [7], the exposure setting for the 16 Hz ELF-PEMF used in the clinical trial (7 min daily for a total of 30 days) is very short. This suggests that modifying the exposure setting bares prospects for optimization. In vitro, the osteogenic potential of the 16 Hz ELF-PEMF was enhanced by prolonging the duration of the daily exposure, with a maximum effect observed with 30 min daily exposure [3]. Further extension of the exposure duration above 1 h even had adverse effects, possibly due to an accumulation of reactive oxygen species (ROS) [8]. In the preceding study, repetitive ELF-PEMF exposures allowed the cells to recover from the formed ROS and to induce anti-oxidative defense mechanisms [8], suggesting even distribution of the daily ELF-PEMF exposure across the day. Notably, the on/off intervals of such an intermittent exposure strategy should not be too short, as this might induce DNA strand breaks as shown, for example, in diploid fibroblasts [9]. For developing an applicable ELF-PEMF exposure strategy, one has to consider that prolonging the daily exposure duration may not only induce local stress but also reduce patient compliance when the ELF-PEMF has to be manually applied for multiple or longer times [10]. One alternative to improve patients’ compliance might be the inclusion of source of the ELF-PEMF within an implant or bone-biomaterial.

One possible mechanism of how ELF-PEMF exposure induces ROS is an increased calcium (Ca^2+^) influx [11]. An increase in intracellular Ca^2+^ by PEMF exposure was shown to promote osteogenesis of bone-marrow-derived mesenchymal stromal cells (MSCs), an effect primarily related to enhanced intracellular Ca^2+^ levels early in the differentiation process and not to increased extracellular Ca^2+^ deposition [12]. This might be due to an increased expression of calcium channels, as has been shown in ELF-PEMF-exposed neuronal cells [13]. One such Ca^2+^ channel, recently identified to play a crucial role in mechanotransduction in bone, is piezo 1. Piezo 1 was defined as a non-selective Ca^2+^ channel with both mechanosensitive [14] and voltage-sensitive abilities [15]. In osteoprogenitor cells, mechanical stimulation was shown to induce expression of *piezo 1* [16,17], which then locates on the plasma membrane to perceive the physical signals generated by the piezoelectric effect of the bone [18]. Piezo 1 transfers biological signals to osteoprogenitor cells through mediating Ca^2+^ influx [19], and its deficiency in osteoprogenitor cells led to the formation of osteoporosis [20]. Considering the important role of piezo 1 in bone homeostasis and its sensitivity towards physical stimuli, this study aimed at exploring if piezo 1 is involved in the osteogenic effects seen in osteoprogenitor cells exposed to 16 Hz ELF-PEMF. Therefore, the 16 Hz ELF-PEMF exposure strategy was optimized for osteogenic effects on SCP-1 cells. As a possible underlying mechanism, *piezo 1* expression and related Ca^2+^ influx were investigated using specific piezo 1 agonists and antagonists.

## 2. Materials and Methods

### 2.1. Culture and Osteogenic Induction of Osteoprogenitor Cells

The human immortalized mesenchymal stromal cell line SCP-1 was kindly provided by Professor Matthias Schieker. The cells were frequently tested to be free of mycoplasma. SCP-1 cells were cultured in α-MEM medium (Gibco, Darmstadt, Germany) supplemented with 5% heat-inactivated fetal bovine serum (FBS) in a water-saturated atmosphere of 5% CO_2_ at 37 °C. Osteogenic differentiation of SCP-1 cells was induced with osteogenic differentiation medium (α-MEM medium supplemented with 1% FBS, 200 μM L-ascorbate-2-phosphate, 5 mM β-glycerol-phosphate, 25 mM 4-(2-hydroxyethyl)-1-piperazineethanesulfonic acid, 1.5 mM CaCl_2_, and 100 nM dexamethasone) [3].

### 2.2. ELF-PEMF Exposure

The ELF-PEMF devices (Somagen®, CE 0482, compliant with EN ISO 13485:2016 + EN ISO 14971:2012) and the ELF-PEMF incubator system were provided by Sachtleben GmbH (Hamburg, Germany). The ELF-PEMFs applied in this study have a fundamental frequency of 16 Hz, which is emitted as groups of pulses (bursts) in sending-pause intervals [3,5]. Two ELF-PEMF exposure strategies were compared in this study: (1) continuous ELF-PEMF exposure: 30 min exposure without break every 24 h; (2) intermittent ELF-PEMF exposure: 10 min exposure every 8 h. An overview of the exposure strategies is provided in Figure 1a.

### 2.3. Sulforhodamine B (SRB) Staining

Total protein content, representing the cell numbers, was quantified by sulforhodamine B (SRB) staining. Briefly, SCP-1 cells were fixed with ice-cold 99% ethanol (−20 °C) for at least 1 h. After washing 3 times with tap water, cells were incubated with SRB staining solution (0.4% SRB in 1% acetic acid, Sigma-Aldrich, Munich, Germany) for 30 min at room temperature. Unbound SRB was removed by washing 4 to 5 times with 1% acetic acid. Bound SRB was resolved with 10 mM unbuffered tris(hydroxymethyl)aminomethane solution (pH 10.5), and OD was quantified photometrically at λ = 565–690 nm with the OMEGA microplate reader (BMG Labtech, Ortenberg, Germany) [5].

### 2.4. Calcein-AM Staining

Cell viability was determined with Calcein-AM staining. SCP-1 cells were washed once with PBS and subsequently incubated with Calcein-AM (2 μM) and Hoechst 33342 (2 μg/mL) in plain medium at 37 °C for 30 min. Images were taken with a fluorescent microscope (EVOS FL, Life Technologies, Darmstadt, Germany) and analyzed with the ImageJ software [3].

### 2.5. Alkaline Phosphatase (AP) Activity

As an early osteogenic marker, AP activity was determined after 2 weeks of differentiation. SCP-1 cells were gently washed with PBS and then incubated with AP reaction buffer (50 mM glycine, 100 mM tris(hydroxymethyl)aminomethane buffer (pH 10.5), 1 mM MgCl_2_, and 2% 4-nitrophenyl-phosphate) at 37 °C for 30 min. Afterward, the formed 4-nitrophenol was quantified photometrically at λ = 405 nm with the OMEGA microplate reader [3,5].

### 2.6. Alizarin Red and Von Kossa Staining

Alizarin red and von Kossa staining were used to evaluate matrix mineralization. Briefly, after 4 weeks of differentiation, SCP-1 cells were fixed with ice-cold 99% ethanol at −20 °C for at least 1 h. Then, fixed SCP-1 cells were gently washed with tap water. For Alizarin red staining, cells were incubated with 0.5% alizarin red solution (pH 4.0) for 30 min at room temperature. Cells were washed 4–5 times to remove unbound alizarin red. Bound Alizarin red was resolved in 10% cetylpyridinium chloride solution and quantified photometrically at λ = 562 nm. For von Kossa staining, cells were incubated with 3% silver nitrate solution for 30 min at room temperature. After washing with tap water, staining was developed with a mixture of 0.5 M sodium carbonate and 10% formaldehyde for 2 min, followed by incubation with 5% sodium thiosulfate buffer for 5 min [3,5].

### 2.7. qRT-PCR

Total RNA of SCP-1 cells was isolated by phenol-chloroform extraction. The resulting RNA pellets were resuspended in diethylpyrocarbonate water. The concentration and purity of the RNAs were determined photometrically, and integrity of the RNA was confirmed by performing agarose gel electrophoresis. Intact RNA was converted into cDNA with a commercial cDNA synthesis kit (Thermo Fisher Scientific, Karlsruhe, Germany) [3]. qRT-PCR was performed using the Green Master Mix High ROX (Genaxxon, Ulm, Germany). Melting curve analysis was performed to ensure specificity of the primers. Relative gene expression of *piezo 1* was calculated with the 2^−ΔΔCT^ method using *EF1α* and *RPL13* as house-keeping genes, which were proven to be most stable with NormFinder and GeNormv3. The primers used in this study are summarized in Table 1.

### 2.8. Measurement of Intracellular Ca^2+^

SCP-1 cells were loaded with 4 µM Fluo-8-AM (Abcam, Berlin, Germany) at 37 °C for 2 h in the dark. Afterward, SCP-1 cells were gently washed 3 times with PBS and incubated in plain medium with 1.5 mM CaCl_2_. Fluorescent intensity at λ_ex_ = 495 nm and λ_em_ = 516 nm was quantified with the OMEGA microplate reader. For time-lapse fluorescent image series, intracellular Ca^2+^ was automatically recorded every 30 s with the EVOS FL fluorescent microscope (Life Technologies, Darmstadt, Germany). Images were analyzed with the time series analyzer V3 plugin of ImageJ software. Yoda 1, Jedi 2, and Dooku 1 (SML1558, SML2532, and SML2397, respectively, Sigma-Aldrich, Munich, Germany) were used in this study as selective piezo 1 agonist or antagonist.

### 2.9. Statistical Analysis

Results are displayed as box plots with individual measurement points or line graphs with error bands. The number of biological (N) and technical (n) replicates for each experiment is given in the figure legends. Due to the sample size, a normal distribution of the data was unable to be assumed; therefore, data were compared by non-parametric Wilcoxon matched-pairs test and one-way or two-way ANOVA, depending on the experimental setting. Statistical analyses were performed with the GraphPad Prism software V.8.0.1 (El Camino Real, USA). *p* < 0.05 was considered statistically significant.

## 3. Results

### 3.1. Three 10 Min ELF-PEMF Exposures per Day More Effectively Promoted Maturation of Osteoprogenitor Cells Than a Single 30 Min ELF-PEMF Exposure per Day

Previously, it was shown that prolonging the daily exposure to the 16 Hz ELF-PEMF enhanced its osteogenic effect, reaching a maximum at 30 min exposure duration [3]. To also investigate effects of the exposure intervals, SCP-1 cells were exposed daily to the 16 Hz ELF-PEMF with a total duration of 30 min per day, either in a continuous (30 min every 24 h) or in an intermittent (10 min every 8 h) setting. Already after 3 days, ELF-PEMF exposure in both settings significantly increased the number of SCP-1 cells (Figure 1b) and their viability (Figure 1c) when compared with the control group without ELF-PEMF exposure. Although having the same accumulated exposure time of 30 min per day, the intermittent exposure strategy exhibited stronger effects than the continuous exposure strategy (46% vs. 24% increase, *p* = 0.0394).

After 14 days, AP activity was significantly increased in ELF-PEMF-exposed SCP-1 cells (Figure 1d). This resulted in an increased mineral deposition in SCP-1 cells after 28 days, as shown by Alizarin red (Figure 1e,f) and von Kossa staining (Figure 1g). The 16 Hz ELF-PEMF effect on the function of SCP-1 cells was again more pronounced in the group with the intermittent exposure strategy than in the group with the continuous exposure strategy (70% vs. 58% increase and 45% vs. 20% increase, respectively).

### 3.2. Intermittent Exposure to 16 Hz ELF-PEMF Induced Expression of Piezo 1 in SCP-1 Cells

Gene expression of *piezo 1* was evaluated by qRT-PCR (Figure 2). With continuing differentiation time, basal *piezo 1* expression increased by approximately 55%. Daily ELF-PEMF exposure enhanced this effect. When compared to the control group, *piezo 1* expression was increased by 32% to 87% with the continuous exposure strategy, and by 112% to 236% with the intermittent exposure strategy.

### 3.3. The 16 Hz ELF-PEMF Exposure Pattern Affected Net-Ca^2+^ Influx in SCP-1 Cells 

To investigate whether the ELF-PEMF-dependent increase in expression of *piezo 1* was displayed in the Ca^2+^ influx into the cells, intracellular Ca^2+^ levels were determined using the fluorescent Ca^2+^ probe Fluo-8. Time-lap fluorescent images exhibited a Ca^2+^ influx in SCP-1 cells exposed to the 16 Hz ELF-PEMF (Figure 3a,b). Automated image analyses revealed an increase in intracellular Ca^2+^ immediately after exposure to the 16 Hz ELF-PEMF (Figure 3c). In SCP-1 cells with 30 min ELF-PEMF exposure (continuous exposure setting), the intracellular Ca^2+^ levels steadily increased until the ELF-PEMF exposure was completed, then slowly declined. The baseline was reached approx. 60 min after the ELF-PEMF exposure was terminated. In line with this observation, the intracellular Ca^2+^ levels in SCP-1 cells with 10 min exposure (intermittent exposure setting) increased only for the duration of the ELF-PEMF exposure. Consequently, the first exposure interval of the intermittent exposure setting was unable to reach the peak Ca^2+^ levels of the continuous exposure setting. However, the second and third exposure intervals of the intermittent exposure setting even exceeded the peak Ca^2+^ levels of the continuous exposure setting. Thus, the net-Ca^2+^ uptake within the first 24 h was significantly higher (4.2-fold, *p* = 0.0009) in SCP-1 cells with 3 × 10 min intermittent exposure when compared to SCP-1 cells with 1 × 30 min continuous exposure to the 16 Hz ELF-PEMF (Figure 3d). This effect persisted for at least 14 days of differentiation, as can be seen by increased Ca^2+^ influx into SCP-1 cells daily exposed to 16 Hz ELF-PEMF. On both day 7 and day 14, the highest Ca^2+^ influx was detected in SCP-1 cells daily exposed to 16 Hz ELF-PEMF with the intermittent exposure setting (Figure 3e,f).

### 3.4. Modulation of Piezo 1-Mediated Ca^2+^ Influx into SCP-1 Cells by Small Chemicals

Piezo 1 agonists Yoda 1 and Jedi 2 were used to prove that piezo 1 plays an important role in mediating Ca^2+^ influx into SCP-1 cells. Therefore, Fluo-8-loaded SCP-1 cells were stimulated with these small chemicals. Both Yoda 1 and Jedi 2 increased the intracellular Ca^2+^ levels in a dose-dependent manner. However, as Yoda 1 more effectively induced Ca^2+^ influx (Figure 4a) than Jedi 2 (1.4-fold higher intracellular Ca^2+^ levels with half the dose/data not shown), this agonist was used for further experiments.

SCP-1 cells stimulated with 2 µM Yoda 1 were first used to investigate the effectivity of the piezo 1 antagonist Dooku 1. Dooku 1 blocked the Ca^2+^ influx caused by Yoda 1 in a dose-dependent manner. With 40 µM Dooku 1, a peak Ca^2+^ influx was no longer detected (Figure 4b).

### 3.5. The 16 Hz ELF-PEMF Exposure-Induced Ca^2+^ Influx Was Abolished by Antagonizing Piezo 1

To explore the relationship between 16 Hz ELF-PEMF exposure induced Ca^2+^ influx and piezo 1, SCP-1 cells were pre-treated with 0, 5, 10, 20, or 40 μM Dooku 1 and then exposed to 16 Hz ELF-PEMF. As reported above, exposure to 16 Hz ELF-PEMF increased intracellular Ca^2+^ levels in SCP-1 cells. Pre-treatment with Dooku 1 dose-dependently reduced Ca^2+^ influx into 16 Hz ELF-PEMF-treated SCP-1 cells. The maximum inhibition capacity was reached with 20 µM Dooku 1 (Figure 4c). A schematic overview on piezo 1 activation and inhibition by small chemicals and 16 Hz ELF-PEMF exposure is given in Figure 4d.

### 3.6. Pharmacological Activation of Piezo 1 Promoted Osteogenic Differentiation of SCP-1 Cells

To prove that piezo 1 mediated Ca^2+^ influx promotes maturation of osteoprogenitor cells, SCP-1 cells were osteogenically differentiated for 28 days in the presence or absence of 2 µM Yoda 1. Stimulation with 2 µM Yoda 1 did not significantly affect cell numbers (Figure 5a). However, the early osteogenic marker AP activity was increased by 2.3-fold (*p* = 0.0391) with 2 µM Yoda 1 treatment after 14 days of differentiation (Figure 5b). This resulted in increased matrix mineralization after 28 days of differentiation (Figure 5c,d).

### 3.7. Pharmacological Inhibition of Piezo 1 Partly Abrogated the Positive Effects of 16 Hz ELF-PEMF Exposure on the Maturation of SCP-1 Cells

SCP-1 cells were osteogenically differentiated for 28 days with or without daily exposure to 16 Hz ELF-PEMF, either in a continuous (30 min every 24 h) or an intermittent (10 min every 8 h) setting. To explore if the positive effects on maturation of osteoprogenitor cells, especially of the intermittent 16 Hz ELF-PEMF exposure, is mediated by Ca^2+^ influx via piezo 1, the cells were additionally treated with the piezo 1 antagonist Dooku 1. A total of 10 µM Dooku 1 was used, as this concentration was sufficient to abolish the 16 Hz ELF-PEMF-dependent Ca^2+^ influx in SCP-1 cells (Figure 4c), but it did not have negative effects on SCP-1 cell numbers (Figure 6a). As observed previously, 16 Hz ELF-PEMF exposure induced AP activity, especially in the intermittent exposure setting. AP activity was almost bisected in the presence of Dooku 1 in all three groups, being significant for the intermittent exposure setting (Figure 6b). In line with the AP activity, matrix mineralization promoted by 16 Hz ELF-PEMF exposure was reduced in the presence of Dooku 1 (Figure 6c–e). In 16 Hz ELF-PEMF-exposed SCP-1 cells in the continuous exposure setting, formation of mineralized matrix was reduced by 20% in the presence of Dooku 1. In 16 Hz ELF-PEMF-exposed SCP-1 cells in the intermittent exposure setting, which formed more mineralized matrix, this effect was more pronounced (25%).

## 4. Discussion

In recent decades, various adjunct treatments to support bone healing have evolved, e.g., mechanical stretching or compression, vibration, and low-intensity pulsed ultrasound or electromagnetic fields. Despite veritable improvement in bone healing [21], their routine clinical use is still limited partly due to the fact that underlying mechanisms are still poorly understood, and application regimes and parameters are not yet optimized for clinical use. In the present study, mechanisms of ELF-PEMF-promoted osteogenesis were investigated, on the basis of the hypothesis that ELF-PEMF exposure utilizes the unique piezoelectric property of bones that can transform mechanical stress into electrical energy to induce cellular processes [22].

Since the pioneer study of Bassett and colleagues in 1974, the therapeutic effects of (ELF-)PEMF have been widely recognized and successfully applied in different clinical studies to improve fracture healing, bone union, or bone quality ever since [7]. Considering that not only the pathology of bone defects but also the ELF-PEMF parameters and exposure strategies varied greatly between the studies, it is not surprising that thus far, no universal ELF-PEMF treatment strategy has been able to be established in the clinical routine. In this study, an ELF-PEMF with a fundamental frequency of 16 Hz was used, which in previous studies was proven to foster the differentiation of osteoprogenitor cells both in vitro and in vivo [5,6]. For this ELF-PEMF prolonging the daily exposure improved its osteogenic effect, however, the maximum effect was reached with 30 min daily exposure [3]. When compared to other studies [7], 30 min daily exposure is still very short; however, further elongation of the daily exposure even damaged the osteoprogenitor cells [3]. This might have been due to an accumulation of ROS formed by this ELF-PEMF, an effect that can be weakened by repetitive ELF-PEMF exposure, which allows the cells to recover and to induce anti-oxidative defense mechanisms [8]. The intermittent ELF-PEMF exposure strategy used in this study is based on this observation. The on/off intervals were chosen with enough recovery time (10 min exposure every 8 h), as shorter on/off intervals (50 Hz, sinusoidal ELF-EMF for 24 h with ≤25 min of recovery) were shown to induce DNA strand breaks in diploid fibroblasts [9]. Interestingly, the amount of DNA strand breaks in diploid fibroblasts with 24 h of continuous exposure to the same ELF-PEMF was significantly lower than for the intermittent exposures with an overall shorter exposure time. Furthermore, it has to be noted that, compared to other cell types, diploid fibroblasts seem to be quite sensitive to stress induced by intermittent ELF-PEMF exposure [23]. This proposes a cell-type-specific response, which might depend on specific parameters, e.g., the fundamental frequency of the underlying ELF-PEMF. While osteoprogenitor cells responded well to ELF-PEMF with fundamental frequencies around 16 Hz [5], monocyte-derived osteoclasts or macrophages responded to ELF-PEMF with higher fundamental frequencies [24].

The ELF-PEMF used in this study has a fundamental frequency close to the EMF induced in the human skeleton by mechanical forces during walking [4]. Thus, by providing the external ELF-PEMF, walk-induced mechanical forces are simulated. Recently, it has been described that such mechanical stimulation induced expression of *piezo 1* in osteoprogenitor cells [16,17]. In line with this observation, 16 Hz ELF-PEMF exposure, especially intermittent exposure, induced expression of *piezo 1* in SCP-1 cells. Being voltage- [15] and mechanosensitive [14], piezo 1 is thought to recognize physical signals mediated by piezoelectric effects in the bone during mechanical load [18]. Through regulating Ca^2+^ influx, piezo 1 controls specific biological effects in osteoprogenitor cells [19]. Exposure to 16 Hz ELF-PEMF induced Ca^2+^ influx in the SCP-1 cells, similar to another study on rat calvaria bone cells [25], and intracellular Ca^2+^ levels increased with exposure duration in the SCP-1 cells. Interestingly, the intermittent exposure strategy that had the same net-exposure duration per day as the continuous exposure strategy resulted in much higher intracellular Ca^2+^ levels. This finding is supported by another study showing increased intracellular Ca^2+^ levels in MSCs repetitively exposed to PEMF [12]. Similar to our findings, the PEMF exposure promoted osteogenic differentiation in these cells. The authors identified this effect to be related to enhanced intracellular Ca^2+^ levels early in the differentiation process and not to increased extracellular Ca^2+^ deposition [12]. In another study, increasing the intracellular Ca^2+^ concentration was even reported to suppress adipogenesis of MSCs [26]. This is of special interest as increased marrow adiposity is associated with progression of osteoporosis [27]. Increased adipogenesis and decreased osteogenesis is also observed in MSCs with reduced *piezo 1* expression due to missing mechanical stimulation [20]. A rapid Ca^2+^ influx and intracellular Ca^2+^ accumulation caused by mechanical strain, and proposedly mediated by piezo 1 [20], are thus required to increase proliferation and maturation of osteoprogenitor cells [28].

Although, Ca^2+^ represents an important secondary messenger for tissue homeostasis and regeneration [29], Ca^2+^ influx in osteoprogenitor cells has to be carefully considered, as it may either exhibit an anabolic effect or lead to apoptosis [30]. For instance, in one study, Ca^2+^ influx was shown to activate PI3K/Akt signaling to enhance anabolic and survival pathways in MSCs [31]. In other studies, Ca^2+^ influx has been reported to activate the Wnt/β-catenin signaling pathway [32], phosphorylate GSK-3β [33], or affect the TGF-β/Smad signaling pathway [34]—all being signaling pathways being involved in osteogenesis. However, Ca^2+^ influx also led to nicotinamide-adenine-dinucleotide/ryanodine-receptor-related apoptosis [35]. Increased Ca^2+^ levels are also associated with an increased formation of ROS due to a Ca^2+^-dependent promotion of ATP synthesis [11]. The resulting increase in the cells’ metabolic state may also explain the enhanced Calcein-AM staining in the 16 Hz ELF-PEMF-exposed SCP-1 cells, especially in the group with the intermittent exposure setting, which showed higher intracellular Ca^2+^ levels than the group with the continuous exposure setting.

## 5. Conclusions

Taken together, this study proved that daily exposure to 16 Hz ELF-PEMF in an intermittent exposure strategy enhanced the positive effects of the conventional continuous exposure strategy in terms of cell viability and maturation of osteoprogenitor cells. It was further shown that this effect is mediated by increased *piezo 1* expression and related Ca^2+^ influx. Thus, the intermittent exposure strategy is a promising way to further optimize the therapeutic effects of the 16 Hz ELF-PEMF regarding fracture healing or even treating osteoporosis. The inclusion of the ELF-PEMF source within an implant or bone biomaterials with automated regulation might be a suitable way to increase compliance when multiple exposures in defined timeframes have to be applied.

## Figures and Tables

**Figure 1 jfb-14-00165-f001:**
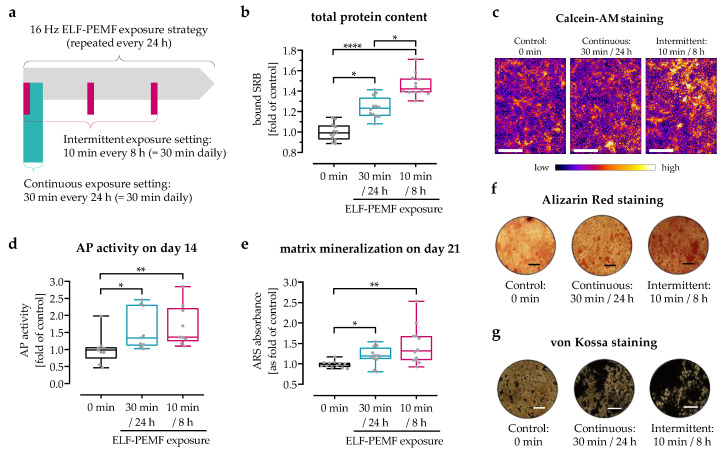
Cell viability and osteogenic differentiation were enhanced by exposure to 16 Hz ELF-PEMF. SCP-1 cells were osteogenically differentiated with or without daily exposure to 16 Hz ELF-PEMF. (**a**) Daily 16 Hz ELF-PEMF exposure was performed either in a continuous setting (30 min every 24 h) or in an intermittent setting (10 min every 8 h). After 3 days of exposure, (**b**) sulforhodamine B (SRB) staining was performed to determine the relative number of cells. (**c**) Calcein-AM staining was used to visualize viable cells; the fluorescent images were pseudo-colored (fire) using ImageJ software; scale bar = 1000 μm. (**d**) AP activity was measured after 14 days of ELF-PEMF exposure. After 28 days of exposure, matrix mineralization was quantified by (**e**) Alizarin red staining and visually confirmed by (**f**) Alizarin red staining and (**g**) von Kossa staining; scale bar = 500 μm. N = 3, n ≥ 3. Data are displayed as box plots with individual data points. Data were analyzed by non-parametric one-way ANOVA followed by Friedman’s multiple comparison test, with * *p* < 0.05, ** *p* < 0.01, and **** *p* < 0.0001.

**Figure 2 jfb-14-00165-f002:**
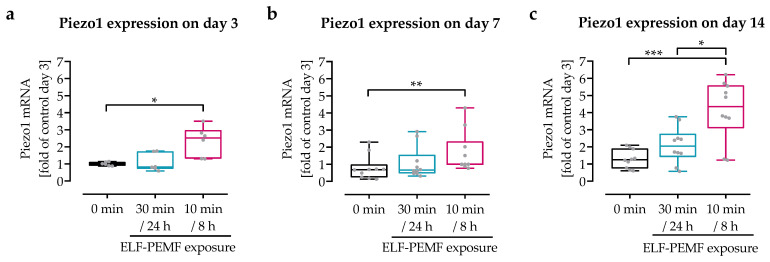
Expression of *piezo 1* increase with differentiation time and 16 Hz ELF-PEMF exposure. SCP-1 cells were osteogenically differentiated for up to 14 days with daily (continuous setting: 30 min every 24 h or intermittent setting: 10 min every 8 h) exposure to the 16 Hz ELF-PEMF. *Piezo 1* expression was determined by qRT-PCR on (**a**) day 3, (**b**) day 7, and (**c**) day 14. Relative gene expression levels were calculated using the 2^−ΔΔCT^ method, with EF1α as the housekeeping gene and the control group of day 14 as the reference. N ≥ 4, n = 2. Data are displayed as box plots with individual data points. Data sets were compared by non-parametric one-way ANOVA followed by Friedman’s multiple comparison test, * *p* < 0.05, ** *p* < 0.01, and *** *p* < 0.001.

**Figure 3 jfb-14-00165-f003:**
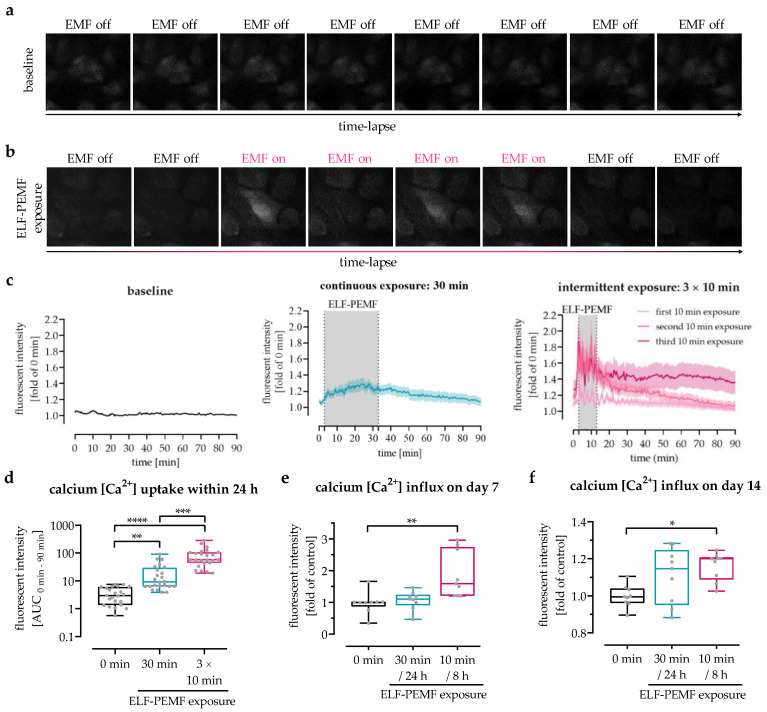
Analysis of 16 Hz ELF-PEMF-dependent Ca^2+^ influx into SCP-1 cells. Intracellular calcium levels were detected using the fluorescent probe Fluo-8. Time-lapse fluorescent image series of Fluo-8-loaded SCP-1 cells (**a**) without or (**b**) with 16 Hz ELF-PEMF exposure. (**c**) Automated image analysis using the time series analyzer V3 plugin of ImageJ software was used to determine intracellular Ca^2+^ levels over time. (**d**) Area under the curve (AUC) was used to determine the net-Ca^2+^ uptake within the first 24 h. Ca^2+^ influx into SCP-1 cells, osteogenically differentiated for (**e**) 7 and (**f**) 14 days with or without daily exposure to 16 Hz ELF-PEMF, was quantified at λ_ex_ = 495 nm and λ_em_ = 516 nm using a microplate reader, 2 min immediately after starting the ELF-PEMF exposure. N = 3, n ≥ 3. Data are displayed as line graphs with error bands or box plots with individual data points. Data were analyzed by non-parametric one-way ANOVA followed by Friedman’s multiple comparison test, * *p* < 0.05, ** *p* < 0.01, *** *p* < 0.001, and **** *p* < 0.0001.

**Figure 4 jfb-14-00165-f004:**
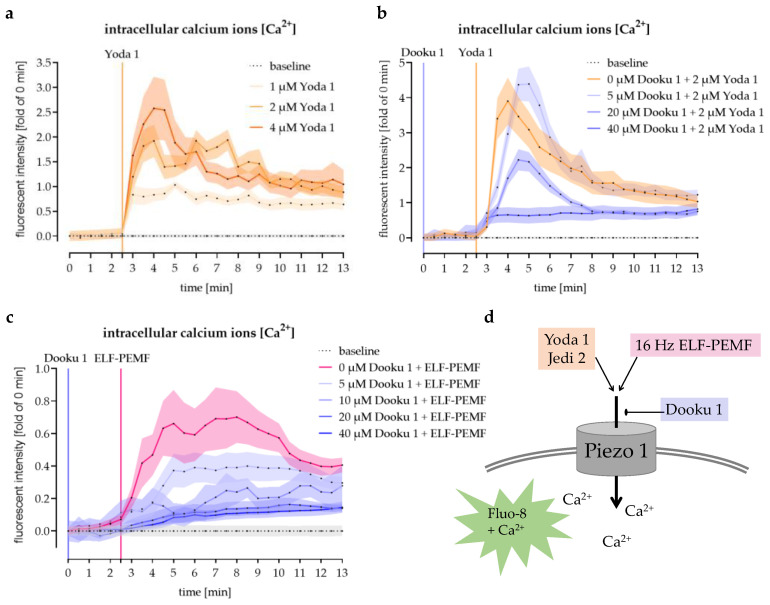
Pharmaceutical modulation of piezo 1-mediated Ca^2+^ influx in SCP-1 cells. Intracellular Ca^2+^ levels were detected using the fluorescent Ca^2+^ probe Fluo-8, delivered to the SCP-1 cells as Fluo-8-AM. Time-lapse fluorescent images were captured (every 30 s) with a fluorescent microscope and analyzed with the time series analyzer V3 plugin of ImageJ software. Intracellular Ca^2+^ levels over time in SCP-1 cells stimulated with (**a**) 0, 1, 2, or 4 μM of the Piezo agonist Yoda 1 or (**b**) 0, 5, 20, or 40 μM of the Piezo antagonist Dooku 1 followed by 2 µM Yoda 1. (**c**) Intracellular Ca^2+^ levels over time in SCP-1 cells stimulated with 0, 5, 10, 20, or 40 μM Dooku 1 followed by 16 Hz ELF-PEMF exposure. (**d**) Schematic overview on piezo 1 activation and inhibition by small chemicals and 16 Hz ELF-PEMF exposure. N = 3, n = 3. Data are displayed as line graphs with error bands.

**Figure 5 jfb-14-00165-f005:**
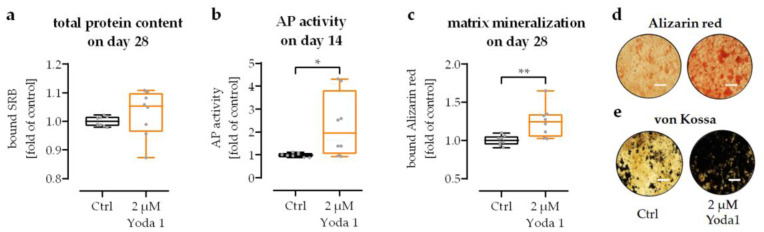
Activation of piezo 1 with Yoda 1 promoted osteogenic differentiation of SCP-1 cells. SCP-1 cells were osteogenically differentiated for 28 days in the presence or absence of the piezo 1 agonist Yoda 1 (2 µM). (**a**) Sulforhodamine B (SRB) staining was performed to determine the relative number of SCP-1 cells on day 28 of differentiation. (**b**) Alkaline phosphatase (AP) activity was determined as an early osteogenic marker at day 14 of differentiation. On day 28, matrix mineralization was quantified by (**c**) bound Alizarin red and visualized by (**d**) Alizarin red and (**e**) von Kossa staining. Scale bar = 500 μm. N = 4, n = 2. Data are displayed as dot plots with individual data points. Groups were compared by Wilcoxon matched-pairs test. * *p* < 0.05 and ** *p* < 0.01.

**Figure 6 jfb-14-00165-f006:**
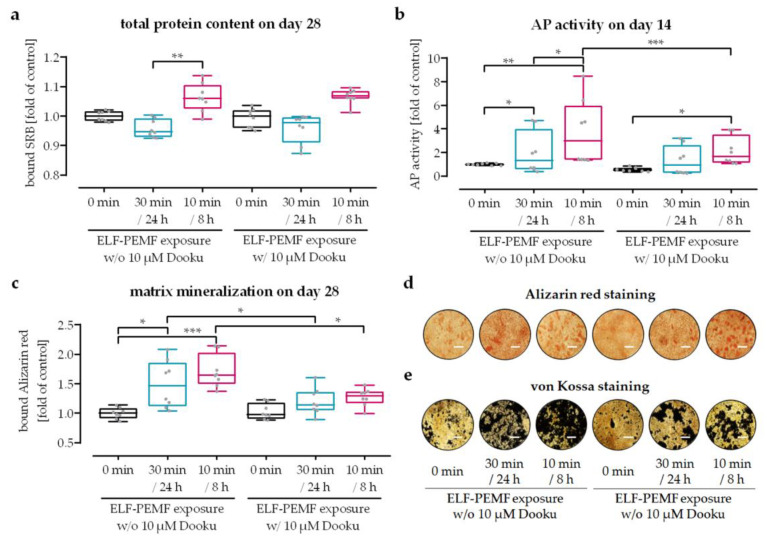
Inhibiting piezo 1 reduced the positive effects of 16 Hz ELF-PEMF exposure on SCP-1 cell maturation. SCP-1 cells were osteogenically differentiated for 28 days with or without daily 16 Hz ELF-PEMF exposure and/or in the presence or absence of 10 µM Dooku 1. Daily 16 Hz ELF-PEMF exposure was performed either in a continuous (30 min every 24 h) or an intermittent (10 min every 8 h) setting. (**a**) Relative cell numbers were determined by sulforhodamine B (SRB) staining on day 28 of differentiation. (**b**) Alkaline phosphatase (AP) activity was determined on day 14 of differentiation. Matrix mineralization was quantified by (**c**) bound Alizarin red and visualized by (**d**) Alizarin red and (**e**) von Kossa staining on day 28 of differentiation. Scale bar = 500 μm. N = 4, n = 2. Data are displayed as box plots with individual data points, and groups were compared by non-parametric two-way ANOVA followed by Tukey’s multiple comparison test. * *p* < 0.05, ** *p* < 0.01, and *** *p* < 0.001.

**Table 1 jfb-14-00165-t001:** Detailed information on primers (designed with primer blast) and the corresponding qPCR conditions.

Target	GenID	Primer	Efficiency	Amplicon	T_A_
*Piezo 1*	NM_001142864.4	For-ACCAACCTCATCAGCGACTT	2.14	212 bp	56 °C
		Rev-AACAGGTATCGGAAGACGGC			
*EF1α*	NM_001402.5	For-CCCCGACACAGTAGCATTTG	1.90	98 bp	56 °C
		Rev-TGACTTTCCATCCCTTGAACC			
*RPL13a*	NM_012423.3	For-AAGTACCAGGCAGTGACAG	2.24	100 bp	56 °C
		Rev-CCTGTTTCCGTAGCCTCATG			

## Data Availability

All data generated and analyzed during this study are included in this published article. The data sets used are available from the corresponding author on reasonable request.

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
