# Peer review of "Intermittent Exposure to a 16 Hz Extremely Low Frequency Pulsed Electromagnetic Field Promotes Osteogenesis In Vitro through Activating Piezo 1-Induced Ca2+ Influx in Osteoprogenitor Cells"

_jfb, 2023, doi:10.3390/jfb14030165_

Round 1

Reviewer 1 Report

In this manuscript, “Intermittent exposure to a 16 Hz extremely low-frequency pulsed electromagnetic field promotes osteogenesis in vitro through activating piezo 1-induced Ca2+ influx in osteoprogenitor cells” by Chen et al. reports the exposure strategy and investigating the underlying mechanisms of a 16 Hz ELF-PEMF. This work is well designed and written. Therefore, I would suggest authors may take a minor revision before publication. Here are the comments and suggestions:

1.        The Y-axis label of the middle Fig. 5c is partial covered.

2.        The response of 2uM Yoda 1 in Fig. 4b is disagreed with that in Fig. 4a.

Author Response

We would like to thank the reviewer for his/her kind word and careful consideration of our manuscript:

In this manuscript, “Intermittent exposure to a 16 Hz extremely low-frequency pulsed electromagnetic field promotes osteogenesis in vitro through activating piezo 1-induced Ca2+ influx in osteoprogenitor cells” by Chen et al. reports the exposure strategy and investigating the underlying mechanisms of a 16 Hz ELF-PEMF. This work is well designed and written. Therefore, I would suggest authors may take a minor revision before publication. Here are the comments and suggestions:

  1. The Y-axis label of the middle Fig. 3c is partial covered.

Answer: We would like to thank the reviewer for this finding and apologize for the mistake. The figure was changed to remove the mentioned overlap.

  1. The response of 2 µM Yoda 1 in Fig. 4b is disagreed with that in Fig. 4a.

Answer: We would like to thank the reviewer for this feedback. This phenomenon is most likely caused by some technical limitations in this assay. Due to the need for constant ELF-PEMF exposure, we could not use the plate scan mode of an automated fluorescent microscope. Thus, different cell passages as well as slight variations in cell density of the SCP-1 cells were used for the different experiments. The fluorescent signal captured with the time-lapse recordings was individually adapted to each run to minimize background signals, which also explains some differences in the fluorescent signals normalized to the baseline fluorescence.

In general, figure 4a shows that 2 µM Yoda 1 is a robust and effective concentration to induce Ca2+ influx in SCP-1 cells. And the main message from figure 4b is the effective inhibition of piezo 1 activity by Dooku 1.

We acknowledge your comment is very important, it points out the technical limitation of using fluorescent probes for measuring Calcium influx. We should find a way to avoid this in future studies. Nevertheless, we are convinced that the observed difference does not affect the conclusion of this study.

Reviewer 2 Report

The manuscript by Yangmengfan Chen et al investigated ELF-EPMF for enhancement of osteogenic differentiation and its mechanism, and found intermittent application promote osteogenic differentiation via peizo1. Overall, the manuscript is well written, the results is convincing. But it is not involved any materials.  Following are some of the comments:

Figure 2 Q-PCR, all y axises said” Fold of control day 14” I think it is wrong, it should be the control of each time points. It does not make sense to use control day 14 for all time points.  If authors want to use one time point to calculate fold change, it should be the day 3, not the day 14. 

Line 263: “Time-laps” spelling error

Figure 5 legend, Authors stated mineralization on day 28, but the label in Figure 5 panel (c) and (d) stated day 21, Please double check which one is correct?

In all figure legends, author put “N” and “n” number, what those “N” and “n” mean need to be clarified.

Figure 6 legends also said “28 days osteogenic differentiation” but figure panel said “21 days”. Please reconcile.

Line 348: “Induced in human the skeleton”. Grammar issue! 

Author Response

We would like to thank the reviewer for his/her kind word and careful consideration of our manuscript:

The manuscript by Yangmengfan Chen et al investigated ELF-EPMF for enhancement of osteogenic differentiation and its mechanism, and found intermittent application promote osteogenic differentiation via peizo1. Overall, the manuscript is well written, the results is convincing. But it is not involved any materials. Following are some of the comments:

  1. Figure 2 Q-PCR, all y axises said” Fold of control day 14” I think it is wrong, it should be the control of each time points. It does not make sense to use control day 14 for all time points. If authors want to use one time point to calculate fold change, it should be day 3, not the day 14.

Answer: We would like to thank the reviewer for his/her comment. We purposely used one time-point for the normalization of the qPCR data in order to show the timely change of the basal piezo 1 expression – we chose day 14 as reference as on that day the piezo 1 expression was highest. As the reviewer suggested to change the reference to day 3, we recalculated the data and changed the figure in the manuscript. However, as expected this did not alter the overall result.

  1. Line 263: “Time-laps” spelling error

Answer: We would like to thank the reviewer for this advice and corrected the spelling as suggested.

  1. Figure 5 legend, Authors stated mineralization on day 28, but the label in Figure 5 panel (c) and (d) stated day 21, Please double check which one is correct?

Figure 6 legends also said “28 days osteogenic differentiation” but figure panel said “21 days”. Please reconcile.

Answer: We would like thank the reviewer for realizing this mistake. The cells have been differentiated for 28 days as stated in the figure legends and text. Therefore, we corrected the labeling of figure 5 and 6 from day 21 to day 28, which is correct.

  1. In all figure legends, author put “N” and “n” number, what those “N” and “n” mean need to be clarified.

Answer: The definition of “N” and “n” is given in the section on statistical analysis (2.9): “The number of biological (N) and technical (n) replicates for each experiment is given in the figure legends”.

  1. Line 348: “Induced in human the skeleton”. Grammar issue!

Answer: We would like to thank the reviewer for his/her comment and corrected the sentence to “…induced in the human skeleton.”